https://doi.org/10.1038/s42003-019-0718-6　　**OPEN**

# Monocytes undergo multi-step differentiation in mice during oral infection by *Toxoplasma gondii*

Aurélie Detavernier[1], Abdulkader Azouz[1], Hussein Shehade[1], Marion Splittgerber[1], Laurye Van Maele[1], Muriel Nguyen[1], Séverine Thomas[1], Younes Achouri[2], David Svec[3], Emilie Calonne[4], François Fuks[4], Guillaume Oldenhove[5] & Stanislas Goriely[1]\*

Monocytes play a major role in the defense against pathogens. They are rapidly mobilized to inflamed sites where they exert both proinflammatory and regulatory effector functions. It is still poorly understood how this dynamic and exceptionally plastic system is controlled at the molecular level. Herein, we evaluated the differentiation process that occurs in Ly6C$^{hi}$ monocytes during oral infection by *Toxoplasma gondii*. Flow cytometry and single-cell analysis revealed distinct activation status and gene expression profiles in the bone marrow, the spleen and the lamina propria of infected mice. We provide further evidence that acquisition of effector functions, such as the capacity to produce interleukin-27, is accompanied by distinct waves of epigenetic programming, highlighting a role for STAT1/IRF1 in the bone marrow and AP-1/NF-κB in the periphery. This work broadens our understanding of the molecular events that occur in vivo during monocyte differentiation in response to inflammatory cues.

[1] Université Libre de Bruxelles, Institute for Medical Immunology and ULB Center for Research in Immunology (U-CRI), Gosselies, Belgium. [2] Université Catholique de Louvain, Institut de Duve, Brussels, Belgium. [3] Institute of Biotechnology, Czech Academy of Science, 252 50 Vestec u prahy, Czech Republic. [4] Université Libre de Bruxelles, Laboratory of Cancer Epigenetics, Brussels, Belgium. [5] Université Libre de Bruxelles, Laboratoire d'Immunobiologie, Gosselies, Belgium. \*email: stgoriel@ulb.ac.be

Monocytes represent a major component of the mononuclear phagocytic system, a specialized system of phagocytic cells localized throughout the body[1]. These cells originate from hematopoietic stem cell-derived progenitors in the bone marrow (BM) that sequentially differentiate into common monocyte progenitors. In mouse, Ly6C$^+$ cells represent the majority of circulating monocytes. They participate to immune homeostasis but are also extremely responsive to inflammatory cues in the context of infections, cardiovascular diseases, autoimmunity, or cancer[2]. Once in the tissues, monocytes complement resident macrophages or dendritic cells (DCs). Depending on the microenvironmental context, they may exert both proinflammatory and regulatory effector functions. It is still poorly understood how this dynamic and exceptionally plastic system is controlled at the molecular level. Selective expression of gene subsets is determined by the activity of distant enhancers during the development of myeloid cells. Lineage-determining transcription factors expressed in progenitors, such as PU.1 or C/EBP act as pioneer factors that establish a unique *cis*-regulatory context[3]. Homeostatic signals delivered by the local microenvironment further shape the enhancer landscape to provide tissue-specific identity and plasticity[4]. Integration of "danger" signals, upon lipopolysaccharide (LPS) stimulation for example, leads to the activation and recruitment of transcription factors (NF-κB, IRFs, and STATs) to these preestablished enhancer/promoter regions, leading to the induction of specific gene expression programs. Nevertheless, some specific stimulus-responsive transcription factors, such as AP-1, IRF8, or STAT1 may also promote the opening of previously inaccessible regions and the acquisition of de novo enhancers that further contribute to cell plasticity[5]. Beyond this first degree of specialization imposed by lineage specifications and microenvironmental factors, it is still not clear how the decision to express specific gene patterns is accomplished. Pioneer transcriptomic studies revealed that within an apparently homogeneous population, distinct patterns of gene activation occur at the single-cell level[6]. However, most of these studies have been conducted under steady-state conditions or on primary macrophages/cell lines cultured in vitro and exposed to a single, well-defined stimulus. While these models are extremely useful to decipher the broad mechanisms involved in the functional specialization of myeloid cells, they probably do not fully recapitulate the sequences of events that occur in vivo upon pathogen encounter. During infection, myeloid cells are exposed to a broad range of microbial/danger-associated stimuli and cytokines that can influence every steps of differentiation. Furthermore, this spectrum of signals is likely to yield strong heterogeneity at the single-cell level in terms of transcriptional and functional output. Herein, we evaluated the differentiation process that occurs in Ly6C$^{hi}$ monocytes during oral infection by *Toxoplasma gondii*. This obligate intracellular parasite rapidly invades the distal half of the small intestine. Parasite burden in the gut peaks 8 days after infection and is associated with acute inflammatory response linked to bacterial translocation[7]. We provide evidence that acquisition of effector functions by Ly6C$^{hi}$ monocytes is accompanied by localization-specific waves of epigenetic programming, dominated by STAT1 in the BM and AP-1/NF-κB in the periphery.

parasite in the lamina propria of the small intestine (SILP) but not in the spleen nor in the BM (Fig. 1a). By day 7, we observed a massive increase in parasitic burden in the SILP and detected its presence in the spleen. Importantly, it remained undetectable in the BM at this timepoint. (Fig. 1a). In this complex setting, we explored the effect of the infection on monocyte differentiation. Ly6C$^{hi}$ monocytes play a major role in parasite killing but also exert regulatory functions[9,10]. These cells were identified as live, CD45$^+$LIN$^-$Ly6G$^-$CD11b$^{hi}$Ly6C$^{hi}$ cells. Under steady-state conditions, they represent a minor proportion of immune cells among splenocytes, in the mesenteric lymph node (MLN) or the SILP (Fig. 1b). In agreement with previous reports[10], the proportion of Ly6C$^{hi}$ monocytes strongly increased in these organs at the peak of infection (day 8). Under these inflammatory conditions, they upregulate CD64 expression (high-affinity IgG receptor FcγRI; Fig. 1c). Induced upon migration of monocytes across the vascular endothelium into tissue, CD64 is linked to transition toward a more activated and/or differentiated cell type[11,12]. Furthermore, it has emerged as a marker for monocyte-derived (mo) DCs[13]. Consistent with this notion, a proportion of these cells acquired CD11c and MHCII expression in the spleen and the MLN (Fig. 1c). Monocytes in the peripheral organs also upregulated costimulatory molecules (CD80, CD86, and CD40). Importantly, some of these phenotypic changes already occurred in the BM as a fraction of Ly6C$^+$ myeloid progenitors expressed CD64 and MHCII at their surface upon *T. gondii* infection. In order to better define the phenotype of these CD11b$^+$Ly6C$^+$ monocytes in these different localizations, we used unbiased clustering *t*-Distribution Stochastic Neighbor Embedding (*t*-SNE) multiparameter analysis of Ly6C, CD64, MHCII, CD11c, CD80, CD86, and CD40 staining. Based on these markers and their behaviors in these different organs, we defined five main cell clusters (Fig. 1d). Clusters 1 and 2 correspond to CD64$^{neg}$MHCII$^{lo}$ monocytes that are mostly found under steady-state conditions. Clusters 3–5 correspond to CD64$^+$MHCII$^{hi}$ monocytes. Cluster 3 is mostly composed of BM monocytes from infected mice but it also harbors cells from the spleen and SILP under steady-state conditions, suggesting that it represents an intermediate differentiation state. Cluster 4 is specifically enriched in the spleen of infected mice and corresponds to cells that also express CD11c and costimulatory markers. Finally, cluster 5 was dominant in the lamina propria and corresponds to cells that express high levels of costimulatory markers but lower levels of CD11c and CD86 as compared to cluster 4. This result supports the notion that differentiation into effector cells is initiated during their development in the BM[14] and provides further evidence that at the peak of the infection, they acquire additional, localization-specific phenotypic features in the periphery. Altogether, this model represents an invaluable situation to assess the differentiation process that occurs in monocytes in vivo prior to egress from the BM, in the spleen under the influence of systemic inflammation and at the site of mucosal infection.

## Results

### Organ-specific phenotype of monocytes during infection.
Oral infection with *T. gondii* cysts (ME-49 strain) induces a Th1-dominated immune response that is required for pathogen control but leads to intestinal immunopathology[8]. We first evaluated the parasite burden in different organs by performing plate-forming unit assay. Four days post infection, we detected the

### Molecular heterogeneity of individual inflammatory monocytes.
To delineate the functional heterogeneity of Ly6C$^{hi}$ monocytes during *T. gondii* infection at the single-cell level, we used an experimental system that allowed us to investigate gene expression of individual monocytes under steady state or in the different organs during infection. Based on our phenotypic analysis, we isolated Ly6C$^+$ monocytes from the BM, the spleen and the SILP under steady-state conditions and Ly6C$^+$CD64$^+$ inflammatory cells from the BM, the spleen (either CD11c$^{hi}$ or $^{low}$ cells), and from the SILP (Gating strategy: Supplementary Fig. 1). We selected genes encoding proteins reported to influence monocyte differentiation or to contribute to effector/regulatory

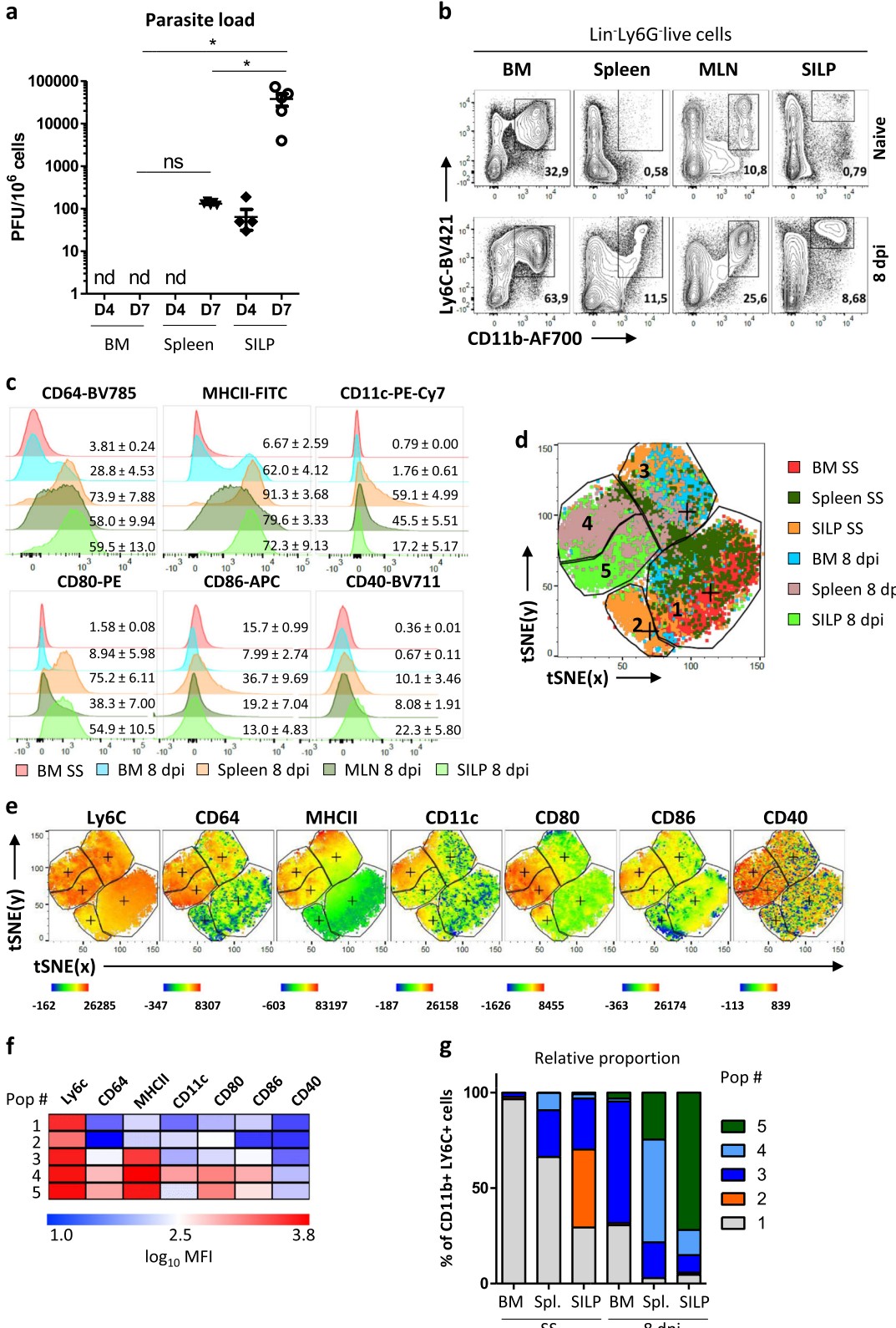

functions, including cytokines, chemokines, surface markers, key metabolic enzymes, and transcriptional regulators (Supplementary Table 1). After excluding failed reactions, we retained expression data from 603 single cells and 79 genes for in-depth analysis (violin plots for individual genes are depicted in Supplementary Fig. 2). Principal component analysis (PCA) revealed that monocytes from the different localizations under steady-state conditions tended to cluster together (Fig. 2a). Upon infection, monocytes isolated from the BM, spleen, and the SILP segregated into distinct clusters. Globally, CD11c$^{low}$ and CD11c$^{high}$ splenic monocytes were indistinguishable. In sharp contrast, cells from the SILP were very different from splenic cells and displayed a considerable degree of heterogeneity. This is compatible with stochastic differences in the activation of regulatory circuits upon

**Fig. 1 Ly6C⁺ monocytes acquire localization-specific phenotype in the course of _T. gondii_ infection.** C57BL/6 wild-type mice were infected perorally with 25 cysts of ME-49 _T. gondii_. **a** Parasite load assessment in the BM, the spleen, and the small intestine lamina propria (SILP) from infected (4 or 7 dpi) mice by plaque forming units (PFU) titration. The experiment was performed once on five mice **b** FACS dot plots of CD11b⁺Ly6C⁺ monocytes from the BM, the spleen, the mesenteric lymph nodes (MLN), and the SILP of naive (steady state) and infected (8 dpi) mice. Data representative of three independent experiments with at least ten mice. **c** Representative histograms showing expression levels of surface markers by CD11b⁺Ly6C⁺ monocytes isolated from the BM, spleen, MLN, or SILP of naive (SS) or infected mice (8 dpi). Numbers indicate percentage of positive cells, expressed as mean ± s.d. from ten individual mice. **d** Unsupervised analysis of single live Lin⁻Ly6G⁻CD11b⁺Ly6C⁺ events from two concatenated mice (one infected and one naive) using nonlinear dimensionality reduction in conjunction with the _t_-Distributed Stochastic Linear Embedding (_t_-SNE) algorithm. **e** Intensity of Ly6C, CD64, MHCII, CD11c, CD80, CD86, and CD40 stainings on the _t_-SNE plots. **f** Median fluorescence intensity (MFI) of surface and activation markers by each of the five _t_-SNE clusters, expressed as log₁₀ MFI. **g** Relative proportion of each population among CD11b⁺ Ly6C⁺ cells in the BM, spleen, and SILP under steady-state conditions or upon infection.

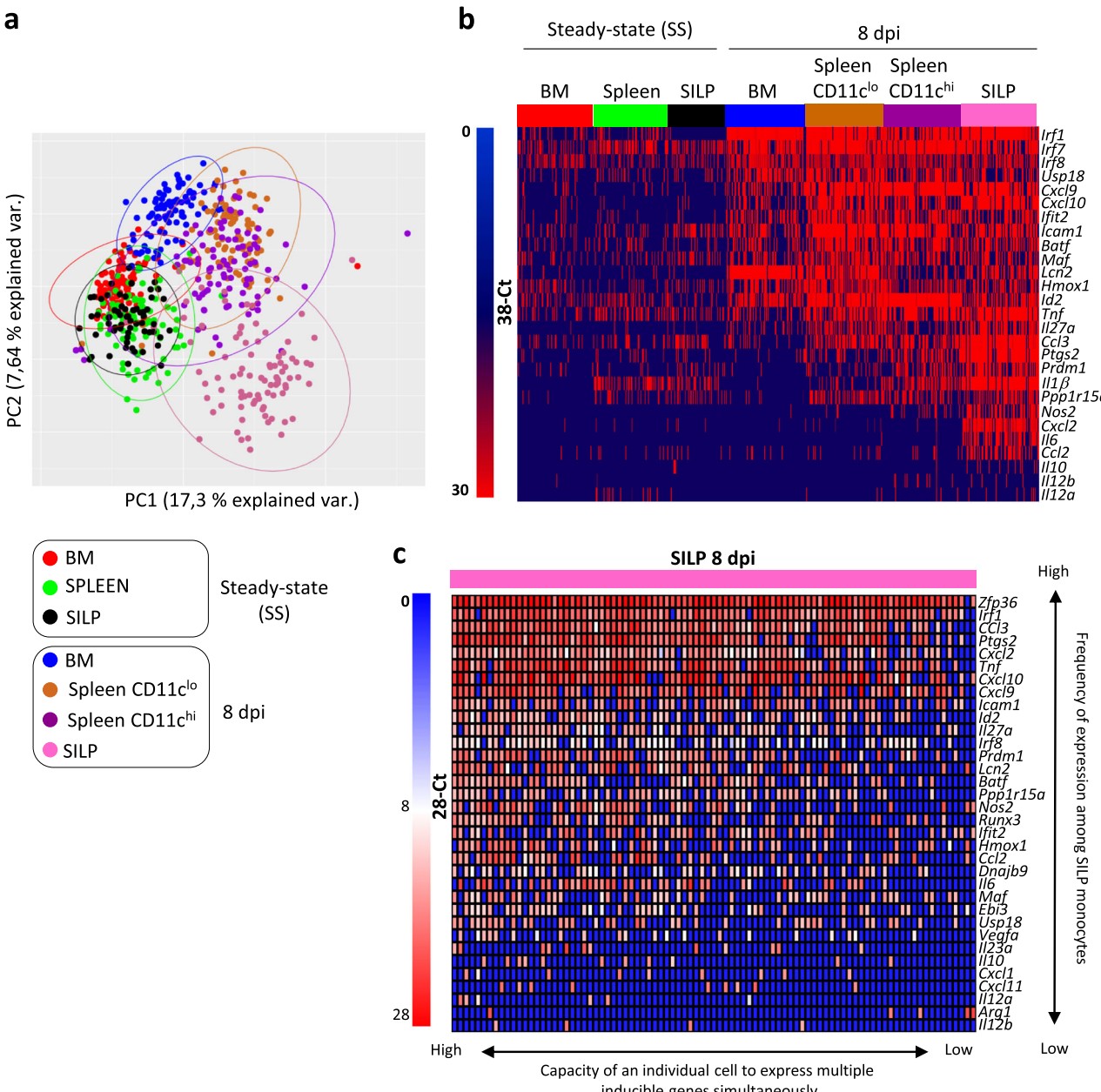

**Fig. 2 Molecular heterogeneity of inflammatory monocytes at the single-cell level.** Transcriptomic profiling of 603 individual monocytes isolated from BM, spleen, and SILP of naive (SS) or infected mice (8 dpi) was performed by single-cell high-throughput qPCR (from a pool of three naïve and seven infected mice). Individual splenic CD11cʰⁱ and CD11cˡᵒʷ monocytes were sorted separately. **a** Principal component analysis of expression of 79 genes by monocytes from BM, spleen (CD11cʰⁱ and ˡᵒʷ), and SILP from naïve (SS) or infected (8 dpi) mice (see the color code). Each dot represents an individual monocyte. **b** Heatmap of expression values (38.7-Ct) of the 27 most differentially expressed genes by monocytes from BM, spleen (CD11cʰⁱ and ˡᵒʷ), and SILP from naive (SS) or infected (8 dpi) mice (see the color code). **c** Heatmap of expression values (28-Ct) of the 34 most differentially expressed genes by monocytes from the SILP of infected mice.

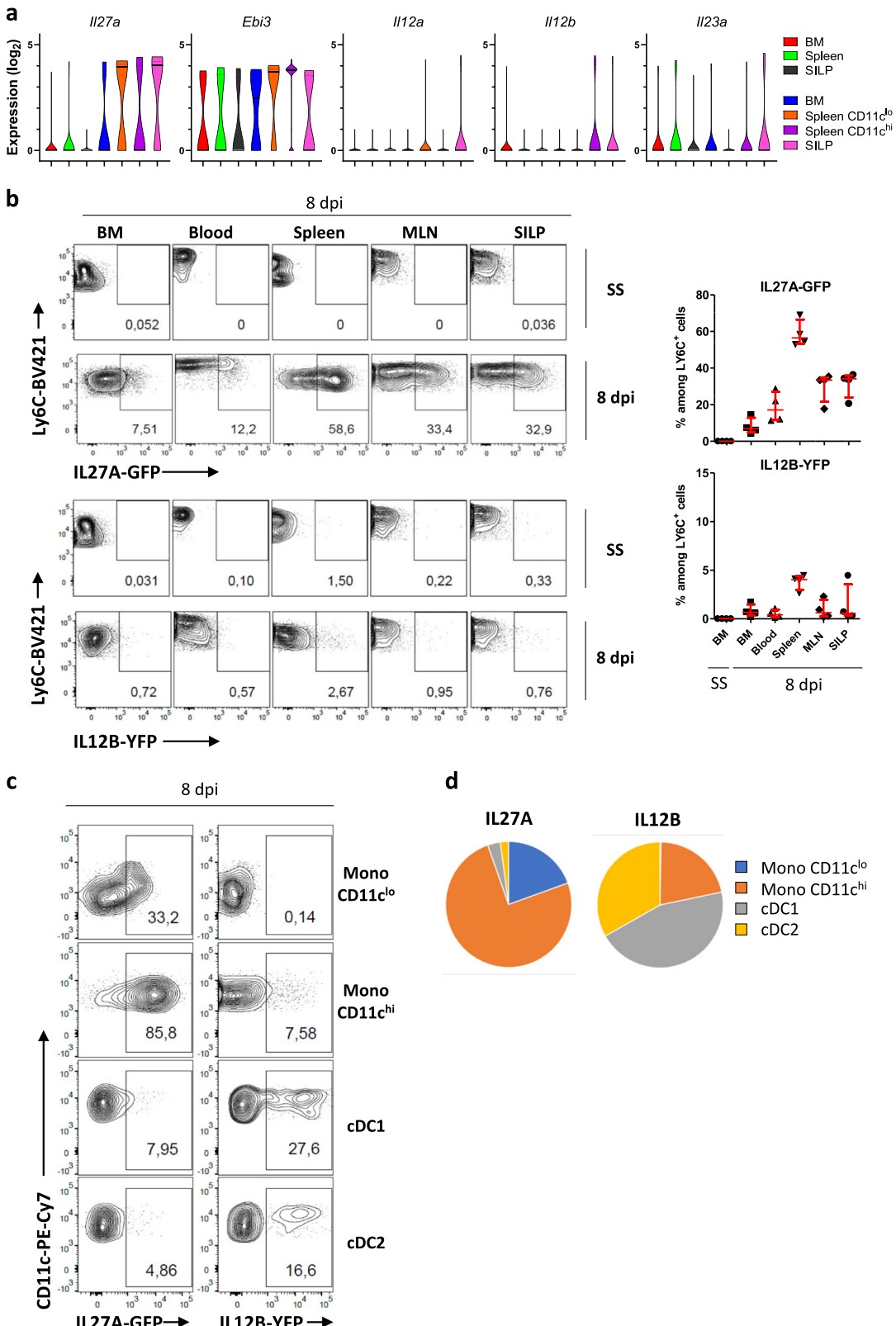

activation[15]. We looked more carefully at the genes that were differentially expressed in these groups (Fig. 2b). Several IFN-dependent genes (such as *Irf1*, *Cxcl9*, or *Ifit2*) were already expressed in BM monocytes from infected mice. This is consistent with the notion that in this model, monocytes are already primed in the BM by IFNγ produced by NK cells[14]. Other genes such as *Tnf*, *Ccl3*, and *Ptgs2* (encoding COX2) were expressed in the spleen and SILP but rarely in the BM. Finally, only cells from the SILP expressed *Nos2*, *Cxcl2*, *Il6*, or *Il10*. Given the important molecular heterogeneity displayed by SILP monocytes from infected mice, we tried to define specific patterns of gene expression. Some genes induced upon *T. gondii* infection, such as *Tnf*, *Cxcl2*, *Cxcl10*, or *Ptgs2* were expressed by a majority of SILP monocytes. This observation suggests that most of these cells have

**Fig. 3 CD11c$^{hi}$Ly6C$^{hi}$ monocytes represent the main source of IL-27 during *T. gondii* infection. a** Violin plots of log$_2$ expression values (38.7-Ct) of the five genes encoding IL-12 (*Il12a* and *Il12b*), IL-23 (*Il23a* and *Il12b*), and IL-27 (*Il27a* and *Ebi3*) by individual monocytes from BM, spleen (CD11c$^{hi}$ and $^{low}$), and SILP from naive (SS) or infected (8 dpi) mice (see the color code). **b–d** IL12B-YFP (Yet40) reporter mice and IL27a-eGFP BAC reporter mice were infected perorally with *T. gondii* cysts. **b** Flow cytometric analysis of CD11b$^+$Ly6C$^{high}$ monocytes from the BM, blood, spleen, mesenteric lymph nodes (MLN), and small intestine lamina propria (SILP) of naive (SS) and infected (8 dpi) mice. Numbers represent the frequency of positive monocytes. Scatter plots indicate the frequency of IL27A-GFP (above) and IL12B-YFP (below) positive monocytes in each organ. Horizontal bars indicate median ± interquartile range and are representative of more than three experiments. Each point represents an individual mouse. **c** Flow cytometric analysis of IL12B-YFP and IL27A-GFP expression by splenic monocytes (CD11c$^{hi}$ and $^{low}$), conventional DC 1 (cDC1) and 2 (cDC2), defined as Lin$^-$MHCII$^{hi}$CD11c$^{hi}$CD64$^-$CD11b$^{low}$CD8$\alpha^+$ and Lin$^-$MHCII$^{hi}$CD11c$^{hi}$CD64$^-$CD11b$^+$CD8$\alpha^{low}$, respectively. Numbers represent the frequency of positive cells. **d** Relative proportion of each population among IL27A-GFP$^+$ and IL12B-YFP$^+$ cells in the spleen of infected mice (means from $n = 4$ per group, representative of three independent experiments).

received activation signals from microbial components or inflammatory cytokines (Fig. 2c). In the same cells, expression of other mediators, such as *Il6*, *Nos2*, and *Ccl2* was clearly bimodal. This could reflect distinct activation status and/or differences in the usage of key regulatory circuits[6]. Finally, expression of some inducible genes, such as *Cxcl1*, *Cxcl11*, or *Il10* was restricted to few cells. This hierarchical organization of inducible genes could be related to temporal or stochastic events, or reflect distinct requirements for long-range contact between regulatory elements that only occur in rare "jackpot" cells[16].

Taken together, these results indicate that the activation status of individual Ly6C$^{hi}$ monocytes is strongly associated with their localization during *T. gondii* infection and that major functional heterogeneity is observed in the lamina propria.

**CD11c$^{hi}$Ly6C$^{hi}$ monocytes represent the main source of interleukin 27.** To further illustrate the specific gene patterns that are observed in monocytes from different localizations, we focused on interleukin (IL)-12 family members. These cytokines play a key role in the control of inflammation. In the context of *T. gondii* infection, IL-12 is critical to control the parasite burden[17]. In contrast, IL-27 displays regulatory functions by promoting the suppressive activity of regulatory T cells[18]. The five genes that encode IL-12, IL-23, and IL-27 are differentially regulated at the epigenetic, transcriptional, and posttranscriptional levels[19]. We evaluated the behavior of these genes as monocytes represent potential sources of these mediators (Fig. 3a). *Il12a* and *Il12b* were expressed at low frequency, only in CD11c$^{hi}$ splenic cells or SILP monocytes from infected mice. *Ebi3* was expressed in a large proportion of cells under steady-state conditions and further increased in CD11c$^{hi}$ splenic monocytes. In contrast, the gene encoding the other subunit of IL-27, *Il27a*, displayed a dynamic behavior with expression induced upon infection in some BM monocytes with a further increase of frequency in the spleen and the SILP. These results suggest that Ly6C$^{hi}$ monocytes might be functionally specialized for the production of IL-27 rather than IL-12/23 upon differentiation induced by inflammatory cues. Previous studies have suggested that DCs rather than macrophages were the main source of IL-27[20]. However, the lack of adequate tools did not allow to formally validate this notion. We therefore developed a *Il27a*-eGFP reporter BAC transgenic mouse to address this specific point. IL-27 was not detected in Ly6C$^+$ monocytes under steady-state conditions (Fig. 3b). Upon *T. gondii* infection, we detected IL-27 in BM and circulating monocytes of infected mice. The frequency of IL-27-positive cells was further increased in monocytes from the spleen, MLN and lamina propria. In parallel experiments, we used IL12B-YFP (Yet40) reporter mice and detected low frequency of IL-12B-producing monocytes. In the spleen, we could demonstrate that CD11c$^{hi}$ monocytes and conventional DCs represent the primary sources of IL-27 and IL-12/23, respectively (Fig. 3c, d). Monocytes were also the main source of IL-27 in the SILP (Supplementary Fig. 3). This result indicates that these populations fulfil

different functions in the course of infection and that monocytes are functionally specialized for high IL-27 production, in particular when they acquire CD11c expression in lymphoid organs or at the site of infection.

**Epigenetic modifications occurring in inflammatory monocytes.** Next, we investigated the molecular processes that underlie acquisition of effector functions. For this purpose, we analyzed epigenomic landscapes of these monocyte subpopulations by ATAC-Seq approaches. This technique allows us to map open chromatin regions throughout the genome. We observed extensive modifications upon *T. gondii* infection, in both BM and splenic monocytes (Fig. 4a). However, pairwise comparisons and PCA revealed that these two subpopulations segregated in very distinct clusters (Fig. 4a, b). In contrast, SILP monocytes clustered with splenic cells, indicating close similarities upon *T. gondii* infection. We also detected very few differentially open regions between CD11c$^{low}$ and CD11c$^{hi}$ splenic monocytes from infected mice. Next, we classified these regions in order to identify specific patterns (Fig. 4c and Supplementary Data 1). Most of the differentially accessible peaks were located in distal regions (more than 2 KB away from transcriptional start sites) rather than in proximal promoters. We used publically available chromatin immunoprecipitation (ChIP)-Seq data from BM-derived DCs to define whether these distal ATAC peaks overlapped with enhancer marks (H3K4me1 or H3K27Ac) or binding sites for pioneer transcription factors, such as PU-1 or C/EBP[21]. About 63% of these peaks overlapped with at least one of these regions, suggesting that a large proportion of these distal regulatory elements corresponds to enhancers that are active in the myeloid lineage (Supplementary Fig 4). In these distal peaks, the most important clusters were composed of regions that were more accessible upon *T. gondii* infection and that were either common to BM, splenic, and SILP cells (cluster I) or found in splenic and SILP cells but not in the BM (cluster II). In sharp contrast, the most important cluster identified in proximal promoter regions encompassed regions that were less accessible upon *Toxoplasma* infection in spleen and SILP cells but not in the BM (cluster VI). Taken together, these data indicate that the enhancer landscape of monocytes is strongly remodelled in the course of *T. gondii* infection. They undergo distinct waves of differentiation in the BM and in the periphery. Importantly, despite important phenotypic and functional differences between monocytes from the spleen and the lamina propria, their global epigenetic profiles were highly similar.

**Two distinct waves of epigenetic reprograming in monocytes.** Next, we focused on the two main clusters of differentially accessible regions (I and II) to define their biological significance. We performed gene ontology analysis using GREAT (genomic regions enrichment of annotations tool)[22]. Some relevant pathways were enriched in both clusters I and II (e.g., regulation of inflammatory response) while cellular response to IFNγ and regulation of JNK cascade were specifically enriched in clusters I and II, respectively

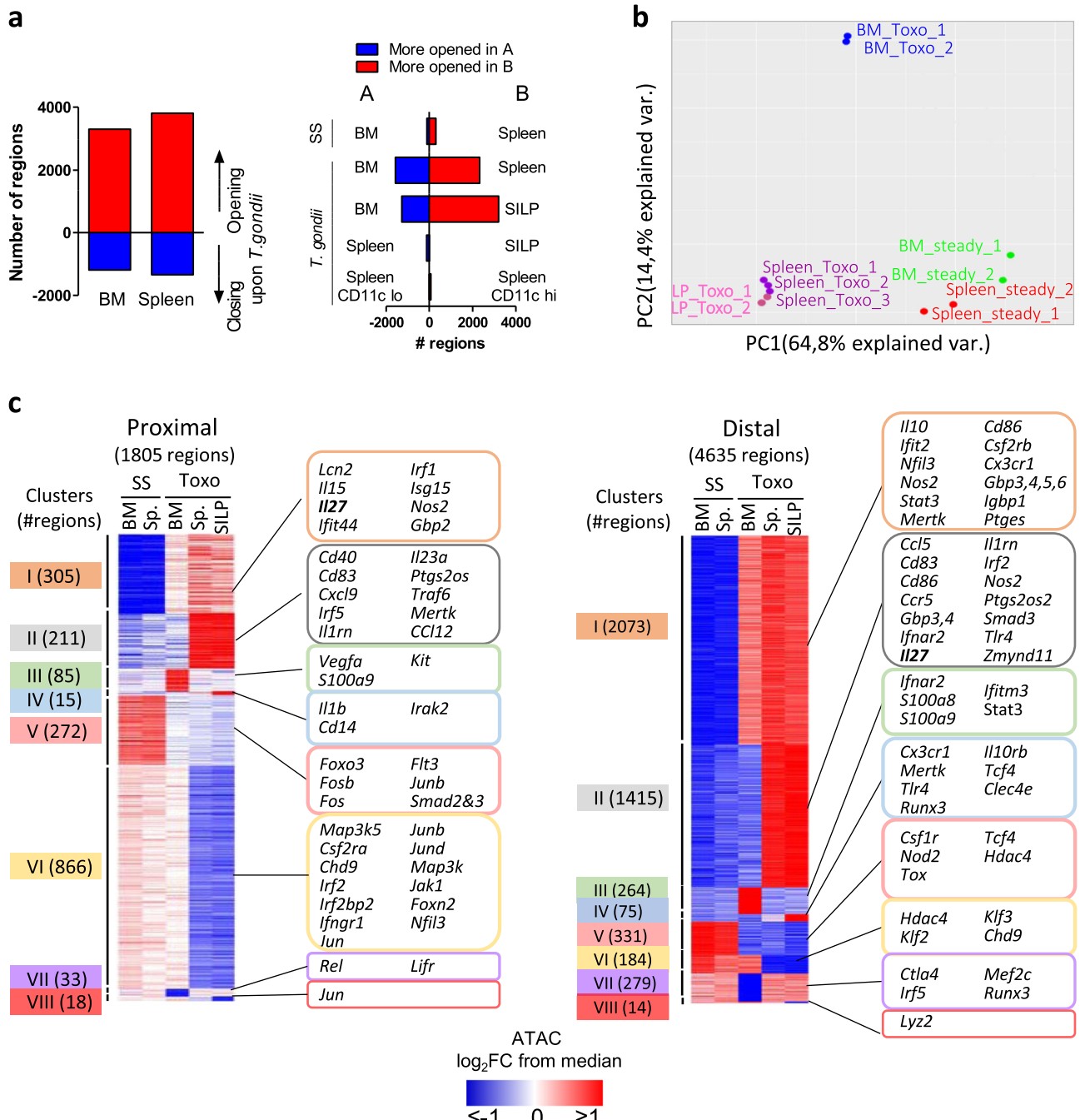

**Fig. 4 During T. gondii infection, monocytes undergo multiple waves of epigenetic modifications. a** Bars indicate the numbers of differentially accessible regions in the BM and the spleen upon *T.gondii* infection (left), and upon pairwise comparisons of the indicated conditions (right). **b** Principal component analysis of ATAC-Seq data from bulk monocytes sorted from BM, spleen, and SILP from naive (SS) or infected mice (8 dpi). Each ATAC-Seq sample was generated from cells obtained from two to three mice. **c** Clustering of proximal (<2 KB from transcriptional start sites) or distal (>2 KB from transcriptional start sites) regions which were more (I–IV) or less (V–VIII) accessible in the BM, spleen, or SILP of infected (8 dpi) compared to naive (SS) mice, respectively. The number of regions is shown in the left margin. Selected genes associated with each cluster are displayed in the right margin. Values are represented as log$_2$ fold-change obtained from median of each single region.

(Fig. 5a). These observations suggest important functional differences between the regulatory regions that are more active already in monocytes from the BM and those that only appear in the spleen and lamina propria. We then scanned for binding motifs at the center of ATAC peaks located in these sets of enhancer regions. Consistent with our pathway analysis, we observed striking differences between clusters I and II: there was a strong enrichment for consensus binding motifs characteristic of STATs and IRFs in

cluster I while AP-1 (JUN/FOS) and NF-κB (REL/RELA) motifs were specifically identified in cluster II (Fig. 5b). We reached similar conclusions for clusters I and II in proximal promoter regions (Supplementary Fig. 5). To further explore the differences between clusters I and II, we used public ChIP-Seq data from BM-derived DCs stimulated or not with LPS and bone marrow-derived macrophages (BMMs) stimulated or not with IFNγ[21,23]. PU.1 and JUNB peaks identified under basal conditions or upon LPS

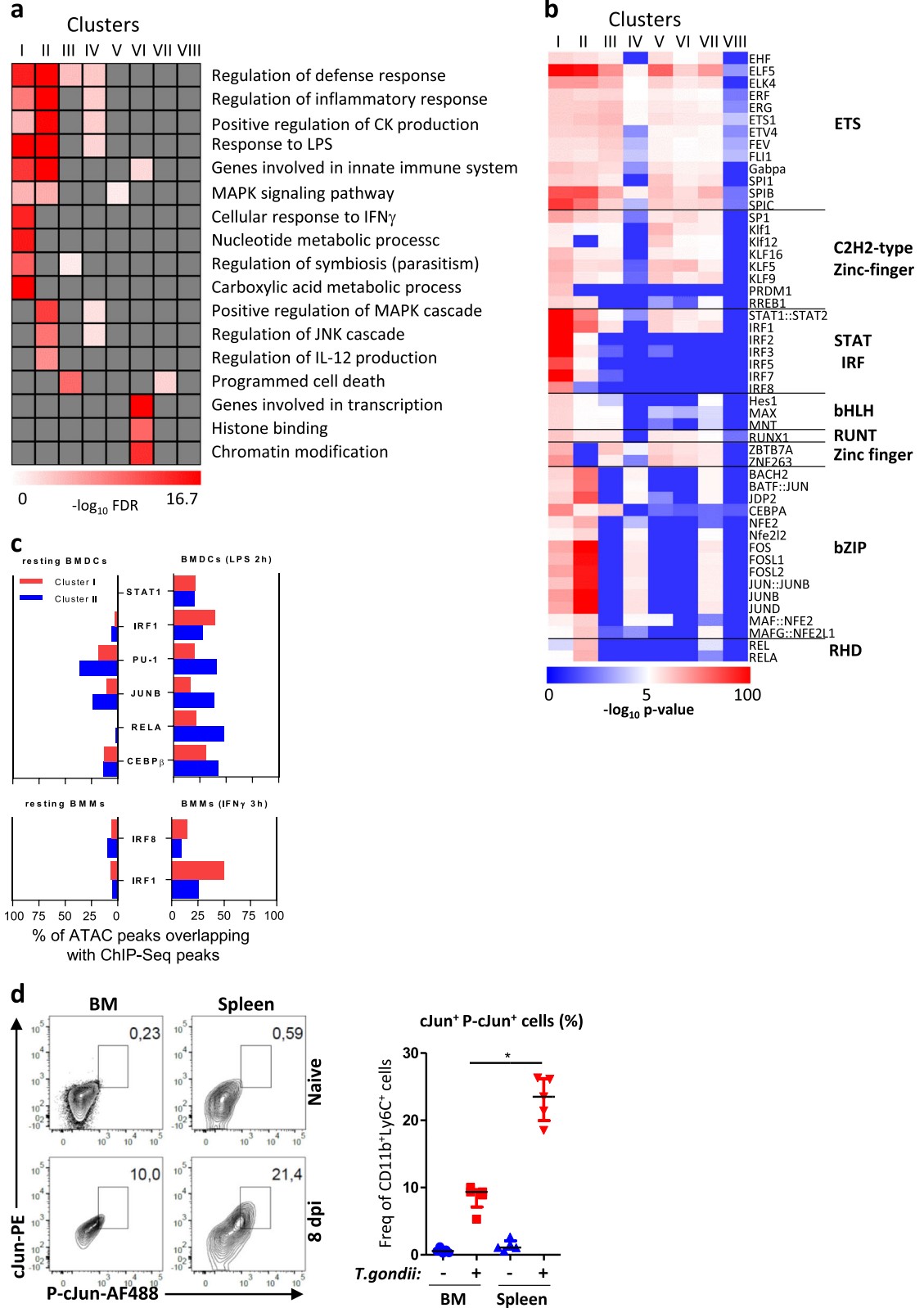

stimulation preferentially overlapped with cluster II regions (Fig. 5c). This was also the case for inducible NF-κB p65 (RELA) and C/EBPβ peaks. In contrast, LPS- and IFNγ-induced IRF1 peaks overlapped to a greater extend with cluster I regions. Notably, the fraction of LPS-induced STAT1 peaks was comparable in both clusters. As a JNK/AP-1 signature was identified in cluster II regions, we assessed c-Jun phosphorylation status in BM and

splenic monocytes by intracellular flow cytometry (Fig. 5d). Consistent with our epigenomic data, we observed that upon *T. gondii* infection, this signaling cascade was more active in splenic cells as compared to their BM counterparts.

Taken together, these data indicate that the first wave of epigenetic reprograming that occurs in the BM involves regions that preferentially harbour STAT and IRF binding sites. In

**Fig. 5 The two waves of epigenetic reprograming are dominated by distinct signaling pathways. a** Selected Gene Ontology pathways enriched in genes associated with clusters I–VIII as defined in Fig. 4c using GREAT with default parameters and presented as −log$_{10}$ of binomial FDR q-value. **b** Motif enrichment analysis of the distal clusters at the centre of overlapping ATAC-Seq peaks using AME and presented as −log$_{10}$ of P values. Transcription factors families are shown in the right margin. **c** Bars indicate the percentage of ATAC peaks overlapping with ChIP-Seq peaks of resting or LPS-stimulated BMDCs or IFNγ-stimulated BMMs, for indicated transcription factors. The numbers of peaks identified for each condition are shown in Supplementary Table 3. **d** FACS analysis of cJun and phospho-cJun expression by CD11b$^+$Ly6C$^+$ monocytes from BM and spleen of naïve (SS) and infected mice (8 dpi). Frequency of cJun and phospho-cJun double-positive monocytes. Horizontal bars indicate median ± interquantile range (each point represents a single mouse). The experiment was performed once on five individual mice. Statistics were calculated using the Mann–Whitney test; *P < 0.05, **P < 0.01.

contrast, the second wave that occurs in the spleen and lamina propria is associated with regions that preferentially harbour binding sites for transcription factors, such as AP-1.

**Critical role of STAT1 in inducing chromatin remodeling.** To specifically assess the cell-intrinsic role of STAT1 in the first wave of chromatin remodeling, we performed mixed BM chimera with Stat1$^{-/-}$ (CD45.2) and WT (CD45.1/CD45.2) cells at a 1/1 ratio. Eight weeks after reconstitution of irradiated CD45.1 congenic mice, we infected the mice with *T. gondii*. (Fig. 6a). Acquisition of inflammatory/maturation markers, such as CD64 or MHCII and CD80 was impaired in absence of STAT1 (Fig. 6b, c). As expected, this was also accompanied by decreased proportion of cells expressing IFN-dependent genes such as *Cxcl9*, *Cxcl10*, *Usp18*, or *Ifit2* as assessed by single-cell quantitative PCR (qPCR; Fig. 6d). Importantly, *Irf1* expression was greatly reduced in absence of STAT1. Consistent with the role of this transcription factor[24], induction of *Il27a* was also abrogated. In contrast, expression of other inflammatory genes, such as *Tnf* or *Lcn2* was not affected. These results indicate that functional programming of monocytes during *T. gondii* infection requires cell intrinsic STAT1-dependent signals.

Next, we performed ATAC-Seq experiments on these cells isolated from the BM or from the spleen of *T. gondii*-infected chimeric mice to define the role of STAT1 at the epigenetic level. We focused our analysis on the accessibility of the regions in the two major clusters of inducible enhancers we had previously defined (see Fig. 4c). We observed that STAT1 was required for the increased accessibility upon infection of up to 60% of the regions that composed cluster I (i.e., the distal regions that are already more active in monocytes from the BM; Fig. 7a). In contrast, <30% of the regions that are part of the cluster II (i.e., the regions that are more accessible in splenic and SILP cells but not in BM monocytes from *T. gondii*-infected mice) were found to be affected by the absence of STAT1. For example, for some genes, such as *Il10*, an enhancer located 3 KB downstream of the Transcription Start Site (TSS) became more accessible in WT BM/Spleen/SILP monocytes from *T. gondii*-infected mice. The increased accessibility of this "cluster I" region was found to be STAT1 dependent (Fig. 7b). For other genes, such as *Il1rn*, cluster II enhancers gained activity in monocytes from the spleen and lamina propria in a STAT1-independent fashion. Finally, other genes, such as *Il27a* follow more complex regulation as their loci harbor both cluster I and cluster II enhancers.

An important proportion of STAT1-dependent regions from cluster I overlapped with STAT1 and/or IRF1 peaks identified in LPS-stimulated BMDCs but not with PU.1, JUNB, C/EBPβ, or RELA peaks (Fig. 7c). As expected, a high proportion of STAT1-dependent regions from cluster II also overlapped with STAT1 and/or IRF1 peaks. Notably, in this case, they also overlapped with PU.1, JUNB, C/EBPβ, and/or RELA peaks, suggesting that the transcriptional network of STAT1/IRF1 in monocytes from the BM or from the spleen of infected mice follows distinct rules. Taken together, these data clearly indicate that STAT1-dependent signals play a dominant role in the initial wave of epigenetic reprogramming that occurs in monocytes from the BM. These signals also participate to

the second wave that occurs in the periphery but by influencing the accessibility of regions that have the capacity to bind other inducible transcription factors, such as AP-1 or NF-κB.

**Discussion**
Our understanding of monocyte development and function has rapidly evolved in the last years with the help of novel experimental approaches, such as fate-mapping or single cell genomics. Under steady-state conditions, Ly6C$^{hi}$ monocytes traffic in the blood and repopulate a proportion of tissue resident macrophages in various organs throughout the body[25]. Alternatively, there is a conversion into Ly6C$^{lo}$ patrolling cells, a process that involves establishment of de novo enhancers. It is now recognized that under pathological situations, alternative differentiation pathways may occur. Furthermore, the origin of monocyte production is influenced by microbial stimuli[26]. Herein, we explored the dynamics of Ly6C$^{hi}$ monocytes during *T. gondii* infection. This complex in vivo situation allowed us to assess different parameters of functional specialization. We show that at the peak of infection, monocytes acquire distinct phenotypic features according to their localization. We observed that a proportion of monocyte precursors in the BM exhibits a phenotypic shift during infection characterized by MHCII and CD64 upregulation and expression of several IFN-dependent genes. This was accompanied by extensive epigenetic reprogramming. Consistent with local NK-derived IFNγ production[14], we show that these processes were strongly dependent on intrinsic STAT1 signaling. Monocytes were heavily recruited to the primary site of infection (small intestine) but also to secondary lymphoid organ, such as the MLN or the spleen. Most of the phenotypic and molecular features acquired upon infection in the BM were maintained in these organs but we observed important additional changes. Monocytes upregulated costimulatory/maturation markers and expressed several effector genes. This was accompanied by a second wave of epigenetic reprogramming that was dominated by AP-1 and NF-κB signatures. This process happened both in splenic and SILP cells. We observe an important degree of heterogeneity in the expression of effector genes at the single-cell level mostly in the lamina propria. One possible interpretation is that it reflects asynchronous activation by local *T. gondii*- or microbiota-derived Pathogen Associated Molecular Patterns (PAMPs) while epigenetic reprogramming results from the influence of soluble factors that broadly impact cells at the population level. However, these data should be analysed with caution as we cannot exclude potential PCR artifacts and batch effects, which are some of the pitfalls of the technological approach we have used. Future work using single-cell ATAC-Seq along with transcriptomic approaches would allow to validate our observations and further address this specific point.

In vitro studies highlighted the role of IFNγ in driving epigenetic changes that determine transcriptional output in response to Toll-Like Receptors (TLR) activation[27]. Our mixed BM experiments with STAT1-deficient cells clearly demonstrate the prominent role of this transcription factor in the first wave of epigenetic remodeling. However, as suggested by motif analysis

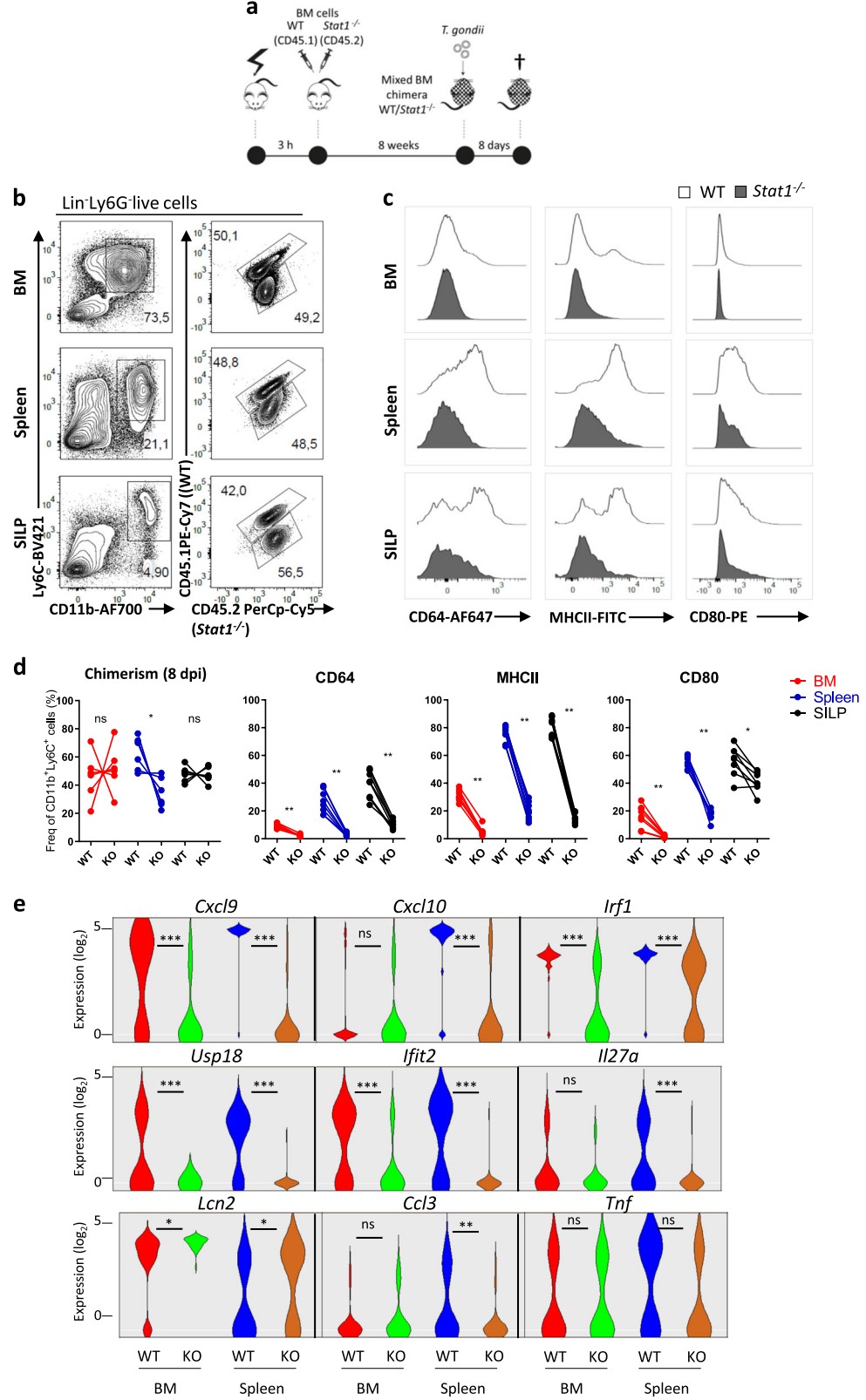

and the overlap with ChIP-Seq data, an important part of this reprogramming could rely on IRF1 rather than STAT1 itself. Indeed, in BMDMs, IRF1 was shown to be required for H3K27Ac deposition to a subset of IFNγ-induced genes[23]. These clusters of elements, bound by PU.1, IRF8, STAT1, and IRF1 represent regulatory hotspots that play a critical role in innate responses.

Our data indicate that these mechanisms are operational in vivo and contribute to monocyte differentiation into effector cells.

The most salient characteristic of the enhancer regions found in cluster II is the enrichment for AP-1 motifs. In fibroblasts, AP-1 binds together with lineage-specific TFs to select the enhancer repertoire through the recruitment of the BAF complex[28]. In

**Fig. 6 Critical role of STAT1 in the acquisition of phenotypic and effector function by inflammatory monocytes. a** Wild type (WT)/*Stat1*−/− mixed BM chimeras were infected perorally with *T. gondii* cysts. **b** Flow cytometric analysis of CD11b+Ly6C+ monocytes from BM, spleen, and SILP, of naive (SS) and infected mice (8 dpi). **c** Representative histograms showing expression levels of surface markers and activation markers by CD11b+Ly6C+ monocytes from BM, spleen, and SILP of naive (SS) and infected mice (8 dpi). **d** Frequency of WT and *Stat1*−/− CD11b+Ly6C+ monocytes in the BM, spleen, and SILP of infected mice (8dpi) and frequency of WT and *Stat1*−/− CD11b+Ly6C+ monocytes expressing CD64, MHCII, or CD80 in these organs (each point represents a single mouse, *n* = 6 mice). Data representative of three independent experiments. Statistics were calculated using Wilcoxon test; *P < 0.05, **P < 0.01. **e** Violin plots of log₂ expression values (28-Ct) of selected genes by individual WT and *Stat1*−/− monocytes from BM and spleen of infected mice, as assessed by single-cell qPCR. Statistics were calculated using Mann–Whitney test; *P < 0.05, **P < 0.01.

BMDCs, JunB is prebound to regulatory elements of LPS-inducible genes, thereby priming them for efficient activation[21]. We propose that similar mechanisms occur in vivo during monocyte differentiation as c-Jun activity was enhanced in splenic monocytes as compared to BM cells. Along this line, TNF was shown to broadly reprogram chromatin state in human macrophages—a finding that was associated with sustained expression of AP-1 proteins[29]. To test the potential role of TNF signaling in vivo, we performed mixed BM experiments with TNFR1-deficient cells. These preliminary experiments revealed that TNF signaling contributed to acquisition of some maturation markers and expression of inflammatory genes (as shown in Supplementary Fig. 6). However, this pathway only marginally accounted for the second wave of chromatin remodeling, suggesting that other systemic soluble factors, such as IL1-related cytokines could be involved.

Inflammatory monocytes may exert both beneficial and detrimental functions in the context of infections and inflammation. On one hand, they exert key antimicrobial functions in the context of *T. gondii*, *Listeria monocytogenes*, or *Leishmania major* infections[9,30–32]. Along the same line, they may promote inflammation in the context of colitis[33]. On the other hand, they were found to display regulatory properties during *T. gondii* infection, mainly through the production of COX-2 and to support *Leishmania donovani* parasite load as they promoted IL-10-producing Th1 cells (referred to as Tr1 cells)[10,34]. We observed that individual cells from the lamina propria may simultaneously express genes encoding inflammatory (e.g., iNOS and TNF, CXCL2, and IL-6) and regulatory (e.g., COX2, HO-1, IL-27, and IL-10) effector molecules, arguing against the existence of separate functional subpopulations among inflammatory monocytes. However, this conclusion is based on the analysis of a set of predefined effector genes in a limited number of cells, a major limitation of the single-cell quantitative reverse transcription PCR approach. Unbiased high-throughput single-cell RNA sequencing approaches on a larger number of cells will help defining further the molecular identities of these cells.

As compared to conventional DCs, we demonstrated that monocytes were functionally specialized to express high levels of IL-27. This cytokine plays a major regulatory role in the context of *T. gondii* infection as it promotes IL-10 and T-bet expression by regulatory T cells thereby limiting detrimental effector responses[18]. Previous works suggested that DC-derived IL-27 represented the major source based on CD11c-Cre-mediated ablation of *Il27*[20]. Our results using a IL-27 reporter mouse system revealed that this could actually reflect the contribution of CD11c-expressing monocytes rather than conventional DCs. Indeed, similar tools were recently used in the context of vaccination and also pinpoint monocytes as an important source of IL-27[35]. The simultaneous expression of *Ebi3* suggests that the heterodimer could be efficiently produced. In contrast, single-cell expression of the other genes encoding IL-12 family members (*Il12a*, *Il12b*, and *Il23a*) was rarely observed in monocytes. Consistent with previous results[17], experiments with IL-12B reporter mice identify conventional DCs as main producing cells. Our epigenetic data indicate that multiple regulatory elements around the *Il27* locus

are progressively gained, first in a STAT1-dependent manner in the BM and then in the periphery, supporting the notion that monocytes are educated to produce this cytokine.

In conclusion, our results indicate that in the context of infection with an intracellular pathogen, monocytes undergo two major steps of epigenetic reprogramming that condition their identity and effector functions. The present work only provides a snapshot of the complexity of the events that occur in vivo but broadens our understanding of the molecular mechanisms that drive functional specialization of monocytes in response to inflammatory cues.

## Methods

**T. gondii infection**. ME49 type II strain was kindly provided by Dr. De Craeye (ISP, Belgium) and was used to produce tissue cysts in C57BL/6 mice, which were inoculated with four cysts by gavage (chronic infection). One month post infection, mice were sacrificed, and the brains were collected. Tissue cysts were counted, and mice were infected by intragastric gavage with 25 cysts (acute infection). Unless specified, mice were sacrificed 8 days post infection.

**Cell collection**. Single-cell suspensions from spleen and MLNs were prepared using standard methods. BM and SILP were processed as previously described[18,36]. Blood was collected from the retro-orbital plexus after anesthesia and resuspended in 20 μl of heparin (Choay) to prevent coagulation. Cells were washed in RPMI 1640 (Lonza) supplemented with fetal calf serum (FCS) 10%, 2 mM L-glutamine, 25 mM Hepes, 1 mM nonessentials amino acids (Lonza), 5 U/ml penicillin, and 5 U/ml streptomycin (Pen-Strep (Lonza)) (complete medium). Cells were then resuspended in 1 ml of ACK (8.29 g/ml NH4Cl, and 1 g/l KHCO3, 37.2 mg/l Na-EDTA (ethylenediaminetetraacetic acid) dissolved in demineralized water) for 1 min at room temperature, except for the blood, which was resuspended in Red blood cell lysis buffer (Beckton Dickinson) following manufacturer's instructions. Cells were washed in RPMI complete medium again.

**Quantitation of parasite tissue loads**. Human primary fibroblasts (75,000 cells) were cultured in 1 ml of DMEM (Lonza) supplemented with FCS 10%, 2 mM L-glutamine, 1 mM nonessentials amino acids (Lonza), 5 U/ml penicillin, and 5 U/ml streptomycin (Pen-Strep (Lonza)), in six wells plates. Forty-eight hours later, 10¹–10⁶ tissue single cells suspensions were added on the fibroblasts. Parasite tissue load was assessed by enumeration of the plaques formed under a microscope, and expressed as the number of plaque forming units per 10⁶ tissue single cells added.

**Flow cytometry**. Single-cell suspensions were washed in phosphate-buffered saline (PBS) and incubated in mAb 2.4G2 (BD Biosciences) and fluorescently conjugated antibodies for cell surface markers in PBS for 20 min at 4 °C in the dark. Dead cells were excluded with Live/Dead Fixable Aqua Dead Cell Stain (Life Technologies). B cells, T cells, and NK cells were gated out using a dump channel corresponding to cells positive for CD3, CD19, and NK1.1 (CD3 145-2C11 APC-Cy7, CD19 1D3 APC-Cy7, and NK1.1 PK136 APC-Cy7, all from BD Biosciences). Ly6G 1A8 PerCP-Cy5 or APC-Cy7 (BD) were used to exclude granulocytes from the analysis. Ly6C AL-21 BV421, CD11b M1/70 AF700, CD11c HL3 PE-Cy7, CD8α 53.6.7 PerCP, CD64 a and b alloantigens X54-5/7.1 AF647 or BV785, CD40 3/23 BV711, CD80 16-10A1 PE, CD86 GL1 APC, and CD45.1 A20 Pe-Cy7 were all purchased from BD Pharmigen as well. MHCII (IA/IE) M5/114.15.2 FITC, c-Kit (CD117) 2B8 FITC, and CD45.2 104 PerCP-Cy5 were purchased from eBiosciences.

For c-Jun and Phospho-c-Jun staining cells were washed in PBS after extracellular staining and fixed in 1 ml paraformaldehyde 4% for 15 min at 37 °C. Cells were then washed in PBS and permeabilized by adding ice-cold methanol 90% while gently vortexing and incubated 30 min on ice in the dark. Cells were washed twice to remove methanol and incubated in PBS BSA 0.5% and C-Jun 60A8 rabbit mAb PE and phospho-c-Jun (Ser73) D47G9 XP rabbit mAb Alexa Fluor 488 (Cell signaling Technology) for 30 min on ice in the dark. Cells were washed and resuspended in PBS BSA 0.5%.

Cell acquisition was performed on a BD LSRII Fortessa. Data were analyzed using FlowJo v10 software (Tree Star) and the *t*-SNE plugin.

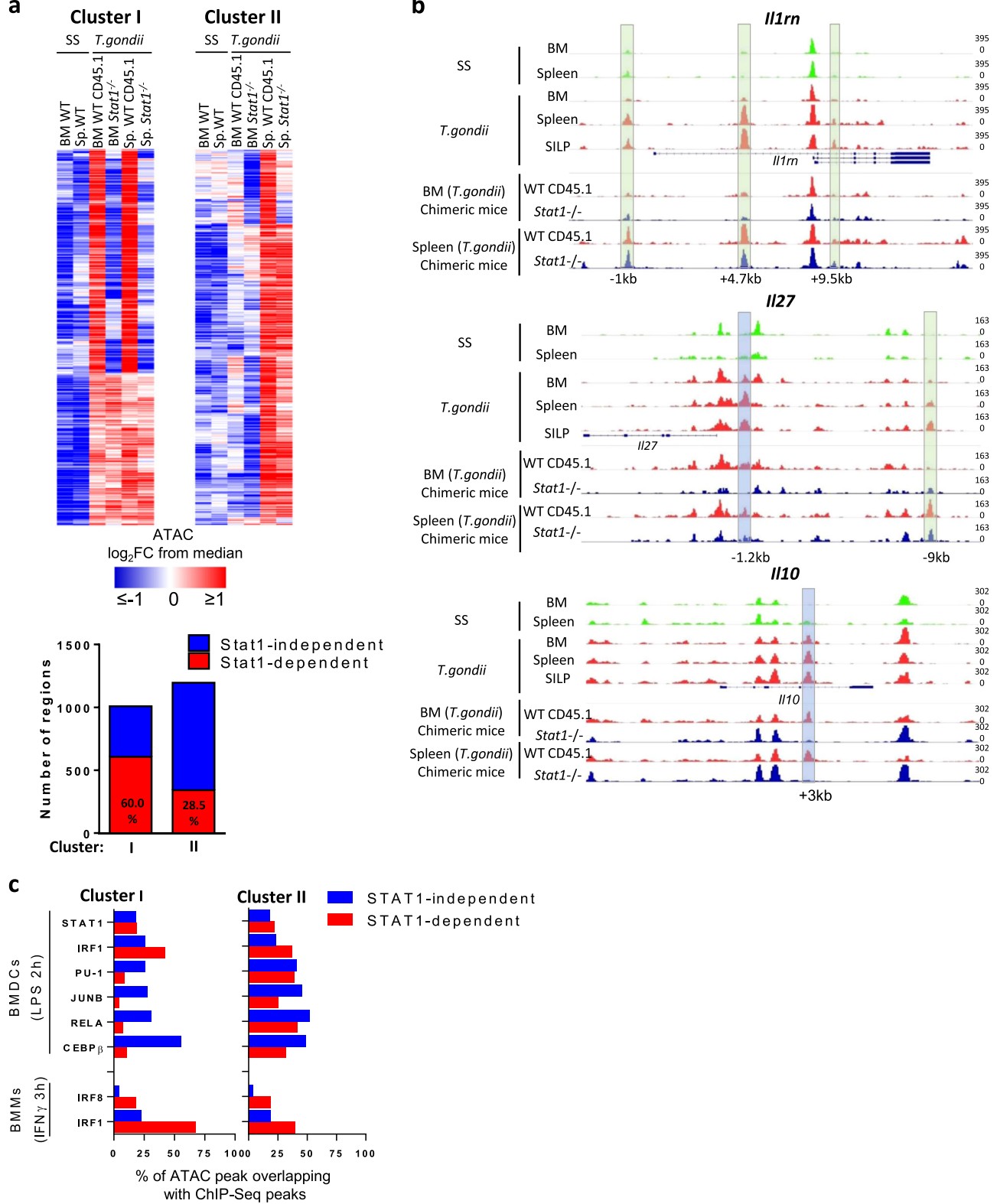

**Fig. 7 Critical role of STAT1 in the induction of chromatin remodeling that occur in the BM.** Wild type (WT)/*Stat1*[−/−] mixed BM chimeras were infected perorally with *T. gondii* cysts as depicted in Fig. 6a. **a** Heatmap showing, among distal regulatory regions from clusters I and II (as defined in Fig. 4), which regions were more or less accessible in CD45.1 WT and CD45.2 *Stat1*[−/−] monocytes from the BM and the spleen of infected mice. Values are represented as log$_2$ fold-change obtained from the median of each single region. Bars indicate the number and the relative proportion of STAT1-dependent (in red) and -independent (in blue) regions among clusters I and II. **b** Representative ATAC-Seq tracks from the indicated population at the loci of *Il1rn*, *Il27a*, and *Il10*. STAT1-dependent and independent peaks are highlighted in blue and green, respectively. The scale represents the fraction of reads in peaks (FRiP). **c** Bars indicating percentage of STAT1-dependent (in red) and -independent (in blue) regions, within clusters I and II enhancer regions overlapping with ChIP-Seq peaks of resting or LPS-stimulated BMDCs and resting and IFNγ-stimulated BMMs, for the indicated transcription factors (as shown in Fig. 5). ATAC samples were generated from cells isolated from a pool of seven chimeric mice.

**Cell sorting**. Cell sorting of monocytes bulks were preceded by a positive selection of CD11b⁺ cells from SILP, spleen, or BM cells resuspended in PBS with 2% FCS (Lonza) and 100 mM EDTA, CD11b MicroBeads, human and mouse (Miltenyi biotec), and Fc receptor-blocking antibodies for 20 min in the dark at 4 °C. Cell separation was performed on the auto-MACS-system according to the manufacturer's recommendations (Miltenyi Biotec). The positive fraction was incubated with Fc receptor-blocking antibodies and a surface staining antibody mix (see Flow cytometry). Single cells and bulks were sorted on a BD FACSAria™ III (see Supplementary Fig. 1 for gating strategy).

**Mice**. C57BL/6 mice were purchased from Envigo. Yet40 (p40-IRES-eYFP, B6.129-$Il12b^{tm11ky}$/J) and Ly5.1 congenic mice on C57BL/6 background were obtained from the Jackson Laboratory $Stat1^{-/-}$ mice on C57BL/6 background were kindly provided by D.E. Levy (New York University School of Medicine, NYC, USA). $Tnfr1^{-/-}$ ($Tnfrsf1a^{tm11mx}$ Jax stock 003242) were purchased from The Jackson Laboratory. For IL27A reporter mice, BAC clones containing $Il27a$ chromosomal regions (BAC ID: RP24-15802) were obtained from the Children's Hospital of Oakland Research Institute BACPAC Resource Center, and then modified so as to insert eGFP gene within first exon of $Il27a$ gene by homologous recombination (Cyagen Biosciences). The resulting modified BAC was then microinjected into the pronuclei of C57BL/6 fertilized eggs and transferred into the oviducts of pseudopregnant females. The pups were screened for the transgene sequence by quantitative real-time PCR using the following oligos: FAM-TTCAAGTCCGCCATGCCCGAATAMRA, 5′-CCACAT GAAGCAGGACTT-3′, and 5′-GGTGCGCTCCTGGACGTA-3′. One of the transgene-bearing pups was subsequently bred with C57BL/6 mice (and backcrossed for at least five generations) to generate $Il27a$ eGFP reporter mice and littermate controls. All experiments were performed on sex- and age-matched (8–12 weeks of age) mice. All animal studies were approved by the Animal Welfare and Ethics Committee of the ULB-IBMM. All experiments were conducted in accordance with the recommended guidelines and regulations.

**Mixed BM chimeras**. Ly5.1 recipient mice were irradiated in two cycles of 600 cGy 3 h apart. Mice were then injected with a suspension of $6.10^6$ of BM cells mixed from $Stat1^{-/-}$ or $Tnfr1^{-/-}$CD45.2, and WT CD45.1.2 mice (50/50) or from WT CD45.2 and WT CD45.1.2 mice (50/50). Eight weeks after BM transplantation, blood was collected (from the retro-orbital plexus after anesthesia), and the degree of chimerism was assessed by measuring CD45.1 and CD45.2 expression by leukocytes using flow cytometry.

**Single-cell qPCR assay**. Measurement of single-cell gene expression levels: Single monocytes were sorted into 96-well plate containing 5 µl Cellulyser Micro lysis buffer (TATAA Biocenter). After cell sorting, samples were centrifuged and stored at −80 °C. Samples were reverse-transcribed into cDNA using the GrandScript cDNA Supermix synthesis kit (TATAA Biocenter; 25 °C for 5 min, 42 °C for 30 min, 85 °C for 5 min, and 4 °C for hold). Samples were then diluted four times by adding 15 µl RNAse/ DNAse-free water, then stored at −20 °C. Ninety samples were centrifuged and a mix containing all primers of interest was added to a final primer concentration of 45 nM. Samples were preamplified using the TATAA Preamp GrandMaster Mix 2 × (TATAA Biocenter) for one denaturation step (95 °C for 3 min) and 20 PCR cycles (each cycle: 95 °C for 20 s, 60 °C for 3 min, and 72 °C for 20 s). The resultant preamplified single-cell cDNA was directly stored at −80 °C until analysis. After thawing on ice, each cDNA sample was diluted four times in RNase–DNAse-free water (Gibco) and 4.5 µl of each sample then combined with 5.5 µl of a mixture containing 5 µl Probe Grand-Master Mix Low ROX (TATAA Biocenter) and 0.5 µl 20X GE sample loading reagent (Fluidigm PN100-7610). Three microliters of combined forward (18 µM), reverse (18 µM), and Taqman probe (5 µM) were prepared for each assay and mixed with 3 µl 2X Assay Loading Reagent (Fluidigm PN100-7611) into another 96 well plate to achieve final primer concentration of 450 nM in qPCR. After priming the BioMark 96.96 IFC chip (Fluidigm) with oil, 5 µl of each assay and 5 µl of each sample were pipetted into their respective inlets on the IFC and the loading was run by choosing; HX: LoadMix (136×). After loading, the IFC chip was placed into the BioMark HD and the thermal protocol was selected (GE 96 × 96 Standard v1.pcl); 40 PCR cycles each composed of one denaturation phase at 95 °C for 15 s and one annealing phase at 60 °C for 60 s were performed. Sequences of the primers and probes are provided in Supplementary Table 1. Among them, $Hprt$ and $Gapdh$ were used as housekeeping genes and individual cells with low Ct values for these genes were excluded from the analysis.

Data collection and analysis for gene expression comparison: Single-cell PCR data were collected and analyzed using Fluidigm Real-time PCR analysis software (version 4.3.1) and Fluidigm Singular Analysis Toolset Software. Heatmap of expression data and violin plots were generated in Singular Analysis Toolset Software. PCA was performed in R using ggplot2 package (H. Wickham. ggplot2: Elegant Graphics for Data Analysis. Springer-Verlag New York, 2016).

**ATAC-Seq**. ATAC-Seq library was prepared as previously described[37]. Briefly, nuclei from 20,000 cells were subject to transposition reaction in 1x TD buffer containing 2.5 µl transposase Nextera enzyme (Nextera DNA sample prep kit, Illumina) and DNA was purified using MinElute PCR Purification Kit (Qiagen). Purified DNA was amplified by PCR using NEBNext High-Fidelity 2 × PCR Master Mix (New England Biolabs) for 10–12 cycles. Amplified libraries were purified

using MinElute PCR Purification Kit (Qiagen) and quality controlled using a Bioanalyzer High-Sensitivity DNA Analysis kit (Agilent). Paired-end sequencing was performed on NovaSeq or NextSeq 500 platforms (Illumina). The list of ATAC-Seq samples is available in Supplementary Table 2.

Mapping and peak calling: Paired-end reads were mapped to mouse genome mm10 with Bowtie2[38,39] using default parameters for paired-end reads. Reads that mapped several regions, or with insufficient mapping quality, were removed with samtools view. We also removed reads located within the blacklist of the ENCODE project[40]. Duplicate reads were removed with MarkDuplicates tools (Picard suite). Peaks were called with MACS2[41] using the following parameters: `-f BAMPE -g mm -q 0.05 --nomodel --call-summits -B -SPMR`.

ATAC-Seq differential analysis: Regions obtained by MACS2 were subjected to differential analysis using SeqMonk 1.43.0 (Mapped Sequence Analysis Tool, Babraham Bioinformatics, http://www.bioinformatics.babraham.ac.uk/projects/ seqmonk/). First, we created an atlas containing all obtained peaks for all the populations using bedtools[42] with a minimum overlap of 1 bp. To identify differentially open/closed regions, a correlation cutoff of 0.9 between similar patterns quantification was applied among all populations. A total of 6441 peaks were identified and seperated into two categories: peaks located in promoters (located within 2 kb around known TSS) and distal peaks (located more than 2 kb from TSS). Each category has eight clusters. For downstream visualization, a scaling factor was calculated using deepTools package[43] to normalize peak intensity to FRiP (fraction of reads in peaks) and generate bigwig files.

Gene ontology analysis: We introduced BED files of each cluster after combining promoters and enhancers to GREAT tool with default parameters[22].

Motif analysis: AME tool from meme-suit[44,45] was used with default parameters to perform motif enrichment in the differentially accessible enhancers regions.

Overlap with ChIP-Seq data: Publicly available ChIP-Seq data of H3K4me1, H3K27Ac, STAT1, IRF1, PU.1, RELA, JUNB, and CEBP (GSE36099)[21] were downloaded and mapped to the mouse genome mm10 using Bowtie2 offered by Galaxy platform[46] in a sensitive local preset parameters. Peaks were called using MACS2 with the following parameters: `--format BAMPE --gsize 2730871774 --bw 300 --mfold 5 50 --qvalue 1e-05` except for STAT1 and JUNB where `--qvalue 1e-04`. Resulted peaks were intersected with clusters I and II, and the percentage of overlap was calculated. For IRF1 and IRF8 ChIP-Seq data from BMMs, we downloaded the BED files (GSE77886) that were overlapped with clusters I and II[23].

**Statistics and reproducibility**. Prism 6.0 was used for statistical analysis. Mann–Whitney test was used to compare two data sets. For all analyses, no data point was excluded. Sample size was determined according to standard practice in the field. All data presented are from representative independent experiments.

**Reporting summary**. Further information on research design is available in the Nature Research Reporting Summary linked to this article.

## Data availability

ATAC-Seq data that support the findings reported in this study have been deposited in the GEO Repository with the accession code GSE129724. Publicly available Chip-Seq datasets from BMDCs (H3K4me1, H3K27Ac, STAT1, IRF1, PU.1, RELA, JUNB and CEBP) and BMMs (IRF1 and IRF8) have been downloaded from GEO Repository with the accession codes GSE36099 and GSE77886, respectively. Source data underlying plots are in Supplementary Data 2 and all other data (if any) are available upon request.

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

## Acknowledgements

This study was supported by the Fonds National de la Recherche Scientifique (FRS-FNRS, Belgium), the WELBIO and the European Regional Development Fund (ERDF) of the Walloon Region (Wallonia-Biomed portfolio, 411132-957270). S.G. is a senior research associate of the FRS-FNRS. A.D. is supported by a grant from the FRIA and the Fonds Rose et Jean Hoguet. D.S. is supported by a Czech Science Foundation grant (18-24753Y), by the institutional support from Czech Academy of Sciences (RVO: 86652036) and by the ERDF projects BIOCEV (CZ.1.05/1.100). We thank Frédéric Libert for NGS sample processing and preliminary analysis, and Thierry Voet and Koen Theunis for giving us the access to the Biomark HD apparatus which was funded by the Hercules Foundation (AKUL/13/41).

## Author contributions

A.D. conducted most of the experiments. H.S., L.V.M. and Y.A. contributed to some experiments; M.N., S.T. and E.C. provided technical help for the experiments. A.A. and M.S. performed bioinformatics analysis. A.D. and A.A. analysed the data and prepared the figures. D.S., G.O. and F.F. provided input for research design and interpretation. S.G. supervised the work and wrote the manuscript. All authors were involved in critically revising the manuscript for important intellectual content. All authors had full access to the data and approved the manuscript before it was submitted by the corresponding author.

## Competing interests

The authors declare no competing interests.
