## [Peer Review File · Communications Biology]

Reviewers' comments:

Reviewer #1 (Remarks to the Author):

The manuscript "Acquisition of Effector Functions by Monocytes during Infection Involves Stepwise Epigenetic Programming" by Detavernier and colleagues provides evidence for a stepwise differentiation of monocytes during *Toxoplasma gondii* infection.

The authors first characterise Ly6C+ monocytes from different organs by FACS and single cell PCR in order to identify tissue- and pathogen-dependent differences. By this they identify monocytes in tissues as the major IL27 (but not IL12B) source during infection. Subsequently ATAC profiling was used to determine epigenetic changes that occur in disease-associated monocytes. STAT1 signatures in the BM as well as AP-1/FOS signatures in the periphery were identified and possibly indicate a stepwise monocytic differentiation process. Indeed, interesting data from well performed mixed-BM chimera experiments support the necessity of STAT1 in the initiation of a disease-dependent monocyte differentiation.

In summary, the paper is well written, the results are of interest and contribute to our understanding of monocyte differentiation during disease. There are only a few points that need to be clarified:

Major points:

1. The manuscript lacks information about number of mice and replicates that were used for experiments. For instance, it is not completely clear if the ATAC data was performed on more than one sample (possibly for instance for BM_SS or SILP_Toxo) or not. At least duplicates should be performed.

2. In some case the authors nicely compare disease phenotypes with the steady state condition (like Figure 1A). However, in the majority of the data, some important steady state conditions are missing. For instance, in Fig. 1C the authors show no depiction of the homeostatic situation. How do monocytes from BM, spleen, MLN and SILP localise in the 2D reduction? Cluster 1 from Fig. 1C is further not homogenous according to CD40 and CD11c (Fig. S1) and parts of this cluster are therefore maybe not of monocytic origin. It is of interest, if these populations already exist in steady state and the experiment should be accordingly adapted.

It is also problematic to show scPCR for SILP cells in Fig. 2A without the homeostatic comparison. These cells appear quite different from BM and spleen cells. This, however, is not completely unexpected. SILP contain even under steady state condition fully differentiated monocyte-derived cells and these cells vary already substantially from BM or splenic monocytes in steady state (see e.g. itgax in immune.org). Therefore, it cannot be concluded if their transcriptomic differences to other monocytic cells as shown in Fig. 2A are due to their localisation or due to disease.

The authors should also show the steady state expression of IL27A/IL12B in Fig. 3B.

3. Due to the mentioned problems that arise from the ATAC (possibly no duplicate) and scPCR data (no homeostatic condition) concerning especially SILP cells, the authors could consider to move these data sets to the supplement and instead focus on the BM and spleen. By this, they could streamline their analysis to these two organs without losing information and make the story easier to read for non-monocyte specialists.

Minor points:

4. Please indicate sequencing depth for the ATAC data. In the IGV tracks depicted in Fig. 7B, are these really 4000-6000 reads for these gene loci or what does the y-axis indicate?

5. The word 'histograms' in the figure legend (Fig. 3B, 7A) is misleading since histograms describe distributions. The words 'quantification', 'analysis' or 'graphic depiction' seems more appropriate.

6. Please add the scale for the FACS data.

7. The authors wrote in the discussion: "In the bone marrow, Ly6Chi monocytes can emerge from either Granulocyte-Monocyte (GMP) or Monocyte/DC (MDP) progenitors, giving rise to cells with different functional potential (Yáñez et al., 2017)." This was shown for disease condition, but not for homeostatic situation. Please indicate accordingly.
8. In the introduction, the authors wrote: "Beyond this first degree of specialisation imposed by lineage specifications and micro-environmental factors, it is still not clear how, at the level of individual cells, the decision to express specific gene patterns is accomplished." Since the authors don't solve the single cell aspect, the part 'at the level of individual cells' can be deleted.
9. The authors wrote "...IRFs in cluster I while AP-1 (JUN/FOS) and NF-kB motifs were specifically identified in cluster II" Please specify NF-kB motifs and provide gene names.
10. Please provide a quantification for Fig. 6A (at least for the supplement).
11. Why did the authors always use SEM for their analysis? SD seems more adequate for these kinds of analysis.

Reviewer #2 (Remarks to the Author):

In the present study by Detavernier et al, the authors investigate the functional heterogeneity of monocytes during acute infection with *Toxoplasma gondii* at different tissue sites, as well as define the epigenetic modifications that lead to this heterogeneity. Specifically, the authors identify CD11c+Ly6C+ monocytes as an important source of IL-27 that begins as early as in the BM, and escalates at peripheral tissues. The authors also describe a step-wise epigenetic change driven by accessibility to STAT1 binding sites in the BM, and accessibility to transcription factors such as AP-1 and NFkB in the periphery. The development of an IL-27 reporter mouse is novel and of significant use to the field. The study is well designed, executed and interpreted appropriately. The authors conclusions are well-supported by the data presented. The findings are important because prior data has described the high likelihood that monocytes responding to inflammation and/or infection will have heterogeneous function within the population. These findings further the field by offering important insight into previously unknown epigenetic mechanisms by which monocyte develop and acquire heterogeneity in the setting of acute infection/inflammation.

In the results for Figure 1, it is inferred from the text that it is the systemic inflammation, not the parasite, that is causing the phenotypic changes to the cells within the bone marrow. Do the authors also observe no infection in the bone marrow as Askenase et al 2015 showed? This point would help the audience to understand how *T. gondii* infection precedes and the impact of the systemic inflammation on sites of immune cell development, which is a critical point for this study.

In the results discussing the Figure 2C, it is difficult to follow the sentence and it is recommended this is reworded for clarity "Some genes encoding cytokines/chemokines..., suggesting that every cells in this infectious context."

The authors refer to Hall et al 2012, to support the finding that IL-27 increases production of Tr1 cells. Tr1 can refer to Th1 cells that produce IL-10, among other regulatory markers. Hall et al showed the effects of IL-27 on Tregs that expressed Tbet, which are typically referred to as Th1-Tregs. Did the authors mean to refer to the previous work showing that IL-27 increased IL-10 in Th1 cells otherwise known as "Tr1"? Since the nomenclature for this field is rather specialized it would be helpful for a more general audience to define what they mean by Tr1 in the text.

Figure 3C shows the reporter for IL27, do the authors find similar expression of IL27 in the SILP populations as well? Similarly, in Figure 5D, is c-Jun phosphorylation similar in the LP?

In the text of the results for Figure 7 need a bit more information for the audience to understand the experiment. In the text referring to Figure 4 for the clusters would help in understanding that reference. Further, it is not explicitly said in the text that the ATAC-seq is from the bone marrow

chimeras. What is the "T. gondii" group in 7B? Is this both STAT1 and WT samples combined? It is unclear how this section is analyzed and the data is compared. A more detailed description would really help the audience.

The figures are correctly referred to in the text of the document but mislabeled on the figures themselves. Two figures labeled Figure 5. Second Figure 5 should be Figure 6. Figure 6 should be Figure 7.

Authors should address grammatical errors present throughout the document.

Number of experimental samples per group and number of repeat experiments need to be included in manuscript.

What was the housekeeping gene used for sc-qPCR analysis?

Point-to-point responses to Reviewers' comments:

Reviewer #1:

The manuscript “Acquisition of Effector Functions by Monocytes during Infection Involves Stepwise Epigenetic Programming” by Detavernier and colleagues provides evidence for a stepwise differentiation of monocytes during *Toxoplasma gondii* infection.

The authors first characterise Ly6C+ monocytes from different organs by FACS and single cell PCR in order to identify tissue- and pathogen-dependent differences. By this they identify monocytes in tissues as the major IL27 (but not IL12B) source during infection. Subsequently ATAC profiling was used to determine epigenetic changes that occur in disease-associated monocytes. STAT1 signatures in the BM as well as AP-1/FOS signatures in the periphery were identified and possibly indicate a stepwise monocytic differentiation process. Indeed, interesting data from well performed mixed-BM chimera experiments support the necessity of STAT1 in the initiation of a disease-dependent monocyte differentiation.

In summary, the paper is well written, the results are of interest and contribute to our understanding of monocyte differentiation during disease. There are only a few points that need to be clarified:

Major points:

- 1. The manuscript lacks information about number of mice and replicates that were used for experiments. For instance, it is not completely clear if the ATAC data was performed on more than one sample (possibly for instance for BM_SS or SILP_Toxo) or not. At least duplicates should be performed.**

As requested, the information (Number of mice and replicates) has now been added in the legends to the figures. Initial ATAC dataset was indeed lacking duplicates for BM_SS and SILP_Toxo. We have now sequenced additional samples and integrated this data to the existing dataset as depicted in Fig 4b.

- 2. A) In some case the authors nicely compare disease phenotypes with the steady state condition (like Figure 1A). However, in the majority of the data, some important steady state conditions are missing. For instance, in Fig. 1C the authors show no depiction of the homeostatic situation. How do monocytes from BM, spleen, MLN and SILP localise in the 2D reduction? Cluster 1 from Fig. 1C is further not homogenous according to CD40 and CD11c (Fig. S1) and parts of this cluster are therefore maybe not of monocytic origin. It is of interest, if these populations already exist in steady state and the experiment should be accordingly adapted.**

Initial tSNE analysis was performed on Ly6C+CD64+ cells that are very rare under steady-state conditions. In order to take into account these conditions, we have therefore re-analyzed the data on total Ly6C+ cells (Fig 1d-g). We had to remove the MLN conditions as too few monocytes

were present under steady state condition in these samples. As suggested by reviewer #1, there is indeed some degree of heterogeneity in the monocytic populations under steady-state conditions, especially in the SILP where CD64+ cells are found. We have adapted the description of these results accordingly.

B) It is also problematic to show scPCR for SILP cells in Fig. 2A without the homeostatic comparison. These cells appear quite different from BM and spleen cells. This, however, is not completely unexpected. SILP contain even under steady state condition fully differentiated monocyte-derived cells and these cells vary already substantially from BM or splenic monocytes in steady state (see e.g. itgax in immune.org). Therefore, it cannot be concluded if their transcriptomic differences to other monocytic cells as shown in Fig. 2A are due to their localisation or due to disease.

We now provide additional scRTqPCR data on SILP monocytes under steady-state conditions. As shown in Fig 2a), they cluster with cells from BM and spleen under steady-state conditions.

C) The authors should also show the steady state expression of IL27A/IL12B in Fig. 3B.

As requested, representative plots for IL12B and IL27A expression in the different organs under steady state conditions is now provided in Fig 3b.

- 3. Due to the mentioned problems that arise from the ATAC (possibly no duplicate) and scPCR data (no homeostatic condition) concerning especially SILP cells, the authors could consider to move these data sets to the supplement and instead focus on the BM and spleen. By this, they could streamline their analysis to these two organs without losing information and make the story easier to read for non-monocyte specialists.**

We hope that the additional data we now provide solve these issues.

Minor points:

- 4. Please indicate sequencing depth for the ATAC data. In the IGV tracks depicted in Fig. 7B, are these really 4000-6000 reads for these gene loci or what does the y-axis indicate?**

Sequencing depths for each ATAC sample are now provided in Table S2. For IGV tracks in Fig 7B, we used FRiP (Fraction Reads in Peaks), representing intensities per million reads and adapted the legend accordingly.

- 5. The word 'histograms' in the figure legend (Fig. 3B, 7A) is misleading since histograms describe distributions. The words 'quantification', 'analysis' or 'graphic depiction' seems more appropriate.**

We changed histograms to "scatter plots" or "bars" .

- 6. Please add the scale for the FACS data.**

We followed this recommendation throughout the figures.

7. **The authors wrote in the discussion: “In the bone marrow, Ly6Chi monocytes can emerge from either Granulocyte-Monocyte (GMP) or Monocyte/DC (MDP) progenitors, giving rise to cells with different functional potential (Yáñez et al., 2017).” This was shown for disease condition, but not for homeostatic situation. Please indicate accordingly.**

We removed this statement as suggested. As this process is enhanced upon exposure to LPS and CpG, respectively, we added the following note later in the discussion:.

“Furthermore, the origin of monocyte production is influenced by microbial stimuli.”

8. **In the introduction, the authors wrote: “Beyond this first degree of specialisation imposed by lineage specifications and micro-environmental factors, it is still not clear how, at the level of individual cells, the decision to express specific gene patterns is accomplished.” Since the authors don’t solve the single cell aspect, the part ‘at the level of individual cells’ can be deleted.**

We followed this recommendation.

9. **The authors wrote “...IRFs in cluster I while AP-1 (JUN/FOS) and NF-kB motifs were specifically identified in cluster II” Please specify NF-kB motifs and provide gene names.**

We added the NF-kB motif (Rel/RelA) as suggested.

497 ATAC peaks among the 1415 regions of cluster II harbored potential Rel or RelA NF-kB consensus sequences (using FIMO, matched p value < 10⁻⁴). Associated genes include *IL12b*, *IL1rn*, *IL1b*, *CCL5*, *Irf1* or *Irf8*. However, we did not add this information to the manuscript.

10. **Please provide a quantification for Fig. 6A (at least for the supplement).**

We now provide individual values for in Fig 6c

11. **Why did the authors always use SEM for their analysis? SD seems more adequate for these kinds of analysis.**

We followed this recommendation and changes Mean-SEM to Median-interquartile range as we did not test the distribution of the values.

Reviewer #2 (Remarks to the Author):

In the present study by Detavernier et al, the authors investigate the functional heterogeneity of monocytes during acute infection with *Toxoplasma gondii* at different tissue sites, as well as define the epigenetic modifications that lead to this heterogeneity. Specifically, the authors identify CD11c+Ly6C+ monocytes as an important source of IL-27 that begins as early as in the BM, and escalates at peripheral tissues. The authors also describe a step-wise epigenetic change driven by accessibility to STAT1 binding sites in the BM, and accessibility to transcription factors such as AP-1 and NFkB in the

periphery. The development of an IL-27 reporter mouse is novel and of significant use to the field. The study is well designed, executed and interpreted appropriately. The authors conclusions are well-supported by the data presented. The findings are important because prior data has described the high likelihood that monocytes responding to inflammation and/or infection will have heterogeneous function within the population. These findings further the field by offering important insight into previously unknown epigenetic mechanisms by which monocyte develop and acquire heterogeneity in the setting of acute infection/inflammation.

12. In the results for Figure 1, it is inferred from the text that it is the systemic inflammation, not the parasite, that is causing the phenotypic changes to the cells within the bone marrow. Do the authors also observe no infection in the bone marrow as Askenase et al 2015 showed? This point would help the audience to understand how *T. gondii* infection precedes and the impact of the systemic inflammation on sites of immune cell development, which is a critical point for this study.

We have now evaluated the parasite burden in the different organs by plate forming unit assay and by qPCR. The first technique was found to be more sensitive and the results are now shown in Fig 1A.

13. In the results discussing the Figure 2C, it is difficult to follow the sentence and it is recommended this is reworded for clarity "Some genes encoding cytokines/chemokines..., suggesting that every cells in this infectious context."

We rephrased this sentence as follows:

"Some genes induced upon *T. gondii*, infection, such as *Tnf*, *Cxcl2*, *Cxcl10* or *Ptgs2* were expressed by a majority of SILP monocytes. This observation suggests that most of these cells have received activation signals from microbial components or inflammatory cytokines"

14. The authors refer to Hall et al 2012, to support the finding that IL-27 increases production of Tr1 cells. Tr1 can refer to Th1 cells that produce IL-10, among other regulatory markers. Hall et al showed the effects of IL-27 on Tregs that expressed Tbet, which are typically referred to as Th1-Tregs. Did the authors mean to refer to the previous work showing that IL-27 increased IL-10 in Th1 cells otherwise known as "Tr1"? Since the nomenclature for this field is rather specialized it would be helpful for a more general audience to define what they mean by Tr1 in the text.

We agree with this comment and rephrased this paragraph as follows:

"This cytokine plays a major regulatory role in the context of *T. gondii* infection as it promotes IL-10 and T-bet expression by regulatory T cells thereby limiting detrimental effector responses".

15. Figure 3C shows the reporter for IL27, do the authors find similar expression of IL27 in the SILP populations as well? Similarly, in Figure 5D, is c-Jun phosphorylation similar in the LP?

As suggested, we now provide IL27 expression data in the SILP populations (Supplementary Fig 3). As c-Jun phosphorylation is extremely labile, the cells need to be fixed as soon as they are extracted from the organs. For these reasons, we only looked in the bone marrow and spleen because collection of live immune cells from the LP is a lengthy process and requires multiple mechanical as well as enzymatical dissociation steps that could lead to artifacts.

- 16. In the text of the results for Figure 7 need a bit more information for the audience to understand the experiment. In the text referring to Figure 4 for the clusters would help in understanding that reference. Further, it is not explicitly said in the text that the ATAC-seq is from the bone marrow chimeras. What is the “T. gondii” group in 7B? Is this both STAT1 and WT samples combined? It is unclear how this section is analyzed and the data is compared. A more detailed description would really help the audience.**

We modified this paragraph as follows:

Next, we performed ATAC-Seq experiments on these cells isolated from the bone marrow or from the spleen of *T. gondii*-infected chimeric mice to define the role of STAT1 at the epigenetic level. We focused our analysis on the accessibility of the regions in the 2 major clusters of inducible enhancers we had previously defined (see Fig 4c). We observed that STAT1 was required for the increased accessibility upon infection of up to 60% of the regions that composed cluster I (*i.e.* the regions that are more active already in monocytes from the bone marrow) (Fig 7A). In contrast, less than 30% of the regions that are part of the cluster II (*i.e.* the regions that are more accessible in splenic and SILP cells but not in BM monocytes from *T. gondii* infected mice) were found to be affected by the absence of STAT1.

In the figure, we also changed the name of WT samples to WT CD45.1 to differentiate the samples that have been generated from normal or chimeric mice.

We also added a schematic figure of the mixed BM experiment in Fig 6a to facilitate the understanding of the set-up.

- 17. The figures are correctly referred to in the text of the document but mislabeled on the figures themselves. Two figures labeled Figure 5. Second Figure 5 should be Figure 6. Figure 6 should be Figure 7.**

We followed this recommendation

- 18. Authors should address grammatical errors present throughout the document.**

We followed this recommendation

- 19. Number of experimental samples per group and number of repeat experiments need to be included in manuscript.**

We now provide this information in the legends to the figures.

20. What was the housekeeping gene used for sc-qPCR analysis?

We used *Gapdh* and *Hprt* as housekeeping genes. We excluded cells with low Ct values for these genes as a first step for the quality control. This is now stated in the M&M section. However, these values were not used for normalization. As indicated in *Technical aspects and recommendations for single-cell qPCR*, (Ståhlberg and Bengtsson, 2010).:

“Reference gene normalization is, however, not applicable to single-cell data, as no RNA (nor protein) is present at constant level due to temporal variations. However, this is rarely a problem as expression data are conveniently reported per cell, which is a natural and intuitive mode of normalization (Bengtsson et al., 2005). Single-cell expression levels are usually reported as relative quantities per cells, but data are also related to absolute transcript numbers as the workflow has negligible losses when based on direct lysis “

Reviewer #3 (Remarks to the Author):

This manuscript examines epigenetic programming of monocytes at several tissue sites (BM, spleen, lamina propria) coupled with single-cell transcriptomic analysis. An interesting picture emerges of an initial priming step in the BM that may be mediated by STAT1 and/or IRF1 (consistent with a previous publication from the Belkaid lab) followed by a second activation step in the periphery that is mediated by AP-1/NF-κB. The authors generate an ATAC-seq data set to complement prior ChIP-seq data sets (with in vitro generated macrophages) and assessed the role of STAT1 using STAT1 KO cells and mixed BM chimeras.

Overall, the manuscript provides interesting insights into tissue-dependent (and presumably time-dependent) epigenetic programming of monocytes in vivo during the course of a model Th1 infection. A few specific points could strengthen the manuscript:

21 The enhancer calls seem to be made solely on the basis of ATAC-seq data, it would be helpful to use published and available data sets on PU.1 and C/EBPα to firm this up

We performed such comparison along with H3K4me1 and H3K27Ac marks obtained from BMDCs to better define the ATAC peaks that were located >2KB from the TSS. This result is now provided as Fig S4 and presented as follows:

“We used publically available ChIP-Seq data from BM-derived DCs to define whether these distal ATAC peaks overlapped with enhancer marks (H3K4me1 or H3K27Ac) or binding sites for pioneer transcription factors such as PU-1 or C/EBP²¹. About 63% of these peaks overlapped with at least one of these regions, suggesting that a large proportion of these distal regulatory elements corresponds to enhancers that are active in the myeloid lineage (Supplementary Fig 4)”.

It shows that a majority of the peaks overlaps with at least of one of these marks/binding sites regions. However, as it does not formally allow us to define these regions as enhancers, we changed this term throughout the manuscript to “distal regulatory regions”.

22. The promoter data looks interesting (Fig. 4C); although the numbers of peaks are not large, an additional analysis could provide interesting insights.

We performed motif analysis on the promoter peaks. We confirmed the STAT1/IRF vs AP1/NFkB preferences for clusters I and II, respectively. These data are provided in Fig S5

23. I could not find the numbers of ATAC-seq replicates, it will be important to have at least 3 and show the PCA plots to help assess the source of variability in experiments.

We now performed biological duplicates (from independent experiments). For some conditions, we have information from additional samples (eg CD11clo or hi cells from the spleen) indicating that experimental variability between the replicates is generally low as shown in the PCA plot (Fig 4a

24. Although I agree with the authors that we are seeing a 'stepwise differentiation process' there is no evidence to directly support this, and this statement should be softened.

We agree with this comment and modified the text accordingly:

In the abstract:

We provide further evidence that ~~this stepwise~~ acquisition of effector functions, such as the capacity to produce interleukin-27, is accompanied by distinct waves of epigenetic programming, highlighting a role for STAT1/IRF1 in the bone marrow and AP-1/NF-κB in the periphery.

They undergo stepwise differentiation processes, first in the bone marrow and then in the periphery

was changed to :

They undergo distinct waves of differentiation in the bone marrow and in the periphery

25. Unfortunately, the authors used an outdated PCR-based strategy for the single cell transcriptomic data – the limitations of this approach including low-ish cell numbers, very small number of genes, and technical/PCR artifacts should be discussed.

We followed this recommendation and added the following paragraphs in the discussion section :

“However, these data should be analysed with caution as we cannot exclude potential PCR artifacts and batch effects, which are some of the pitfalls of the technological approach we have used. Future work using single-cell ATAC-Seq along with transcriptomic approaches would allow to validate our observations and further address this specific point”.

And

“However, this conclusion is based on the analysis of a set of predefined effector genes in a limited number of cells, a major limitation of the single-cell RTqPCR approach. Unbiased high-throughput scRNA-seq approaches on a larger number of cells will help defining further the molecular identities of these cells”.

REVIEWERS' COMMENTS:

Reviewer #1 (Remarks to the Author):

The authors performed a careful revision and sufficiently answered all my previous questions. I therefore recommend publication.

Reviewer #2 (Remarks to the Author):

The authors addressed all comments.

Reviewer #3 (Remarks to the Author):

The authors have addressed my comments and improved the manuscript.